# The Relationship of Left Ventricular Diastolic Dysfunction and Asymmetrical Dimethylarginine as a Biomarker of Endothelial Dysfunction with Cardiovascular Risk Assessed by Systematic Coronary Risk Evaluation2 Algorithm and Heart Failure—A Cross-Sectional Study

**DOI:** 10.3390/ijerph20054433

**Published:** 2023-03-01

**Authors:** Livija Sušić, Lana Maričić, Ines Šahinović, Kristina Kralik, Lucija Klobučar, Mateja Ćosić, Tihomir Sušić, Josip Vincelj, Antonio Burić, Marko Burić, Matea Lukić

**Affiliations:** 1Department of Specialist-Consultative Health Care, Health Center of Osijek-Baranja County, 31000 Osijek, Croatia; 2Faculty of Medicine, University “Josip Juraj Strossmayer”, 31000 Osijek, Croatia; 3Department of Cardiology, University Hospital Center Osijek, 31000 Osijek, Croatia; 4Department of Clinical Laboratory Diagnostics, University Hospital Center Osijek, 31000 Osijek, Croatia; 5The Information Institute Osijek, 31000 Osijek, Croatia; 6NovaMed Health Center Zagreb, 10000 Zagreb, Croatia; 7Department of Radiology, Health Center of Osijek-Baranja County, 31000 Osijek, Croatia

**Keywords:** amino terminal pro-brain natriuretic peptide, asymmetric dimethylarginine, cardiovascular diseases, diastole, drug therapy, endothelium, heart failure, middle aged, risk assessment

## Abstract

**Background:** Cardiovascular (CV) risk factors, causing endothelial dysfunction (ED) and left ventricular diastolic dysfunction (LVDD), contribute to an increased risk of heart failure (HF). The aim of this study was to determine the relationship between the occurrence of LVDD and ED with CV risk assessed by the Systematic Coronary Risk Evaluation2 (SCORE2) algorithm and HF. **Methods:** In the period from November 2019 to May 2022, a cross-sectional study that included 178 middle-aged adults was conducted. Transthoracic echocardiography (TTE) was used to assess left ventricular (LV) diastolic and systolic function. ED was assessed using the plasma values of asymmetric dimethylarginine (ADMA) and was determined using the ELISA method. **Results:** The majority of subjects with LVDD grades 2 and 3 had high/very high SCORE2, developed HF and all were taking medication (*p* < 0.001). They also had significantly lowest plasma ADMA values (*p* < 0.001). We found that the reduction of ADMA concentration is influenced by certain groups of drugs, or more significantly, by their combinations (*p* < 0.001). **Conclusions:** In our study, we confirmed a positive correlation between LVDD, HF and SCORE2 severity. The results showed a negative correlation between the biomarkers of ED, LVDD severity, HF, and SCORE2, which we believe is due to medication effects.

## 1. Introduction

According to the World Health Organization’s (WHO) 2021 report, cardiovascular diseases (CVD) are still the leading cause of death in the world [1]. Despite significant progress in diagnosis and therapy, we continue to see a high prevalence of coronary heart disease (CHD) [2], which could be caused by the formation and accumulation of atherosclerosis or by vasoconstriction, both due to impaired endothelial function. Heart failure (HF) is another prevalent form of CVD which is continuously rising [3]. It occurs as a result of increased intracardiac pressure and/or inadequate cardiac output due to structural and/or functional abnormalities of the heart, the etiology of which varies according to age and geographical location. In adult populations that have Western lifestyles and in developed countries, CHD and hypertension are dominant factors. Characteristic symptoms of HF are dyspnea, ankle swelling, and fatigue, which may be accompanied by signs of fluid accumulation. In addition to the presence of characteristic symptoms and signs, in order to confirm the diagnosis of HF it is necessary to determine cardiovascular (CV) risk factors and the plasma concentration of natriuretic peptides (NP), as well performing an ECG and TTE. When confirming the diagnosis of HF, it is also necessary to determine the phenotype and its severity. The severity of HF is determined by the New York Heart Association (NYHA) and includes four classes: with NYHA class I representing the mildest, almost asymptomatic form, while NYHA class IV represents the most severe form, in which symptoms occur even at rest. The HF phenotype is determined based on the measurement of the systolic function of the left ventricle (LV) using TTE, the so-called left ventricular ejection fraction (LVEF), whereby HF is divided into HF with reduced (HFrEF), mildly reduced (HFmrEF), and preserved ejection fraction (HFpEF) [4]. In all these phenotypes, a certain degree of left ventricular diastolic dysfunction (LVDD) is always present. LVDD, the earliest change common to many CVDs and CV risk factors [5,6,7], is defined as the inability of the LV to receive blood from the left atrium (LA) and pulmonary veins (PV) during diastole without an increase in filling pressure. According to the severity, LVDD can be divided into three grades: grade 1 is characterized by the slowing down of ventricular relaxation without significant hemodynamic consequences, grade 2 is characterized by an increase in LV filling pressures and LA dilatation, and grade 3 is characterized through the development of PV congestion. Although LV catheterization is the gold standard for the diagnosis of LVDD, the most common method for its evaluation in clinical practice is TTE [8]. In order to reduce the burden of atherosclerotic CVDs, the European Society of Cardiology (ESC) proposed new algorithm for estimating the 10-year risk of fatal and non-fatal CV events called the Systematic Coronary Risk Evaluation2 (SCORE2) [9]. Based on the estimated SCORE2, individuals are divided into three risk groups, namely low to moderate, high, and very high CV risk, which further determines the time of specific therapy introduction and the target values of systolic blood pressure (SBP) and low density lipoprotein cholesterol (LDL-C) plasma concentration. Given that SCORE2 does not cover all known CV risk factors, some patients remain underestimated; so in recent decades, scientists have been focused on researching the molecular connection between CV risk factors, CVDs, CHD, and HF. The universal pathophysiological change that connects all entities is endothelial dysfunction (ED) [10]. ED is a disorder responsible for reducing the bioavailability of nitric oxide (NO), which leads to the development of atherosclerosis directly and, via the NO-soluble guanylate cyclase-protein kinase G (sGC-PKG) signaling pathway, to disturbed cardiomyocyte remodeling indirectly [11,12,13]. Although the flow-mediated dilatation (FMD) of the brachial artery is still considered the gold standard for the assessment of systematic ED, recently, biomarkers have been increasingly used for this purpose [12,14]. One of the popular biomarker is asymmetrical dimethylarginine (ADMA), whose increased plasma concentration is associated with major CV events and mortality, even in subjects without previously proven CVD [15]. ADMA is an endogenous inhibitor of NO synthase (NOS), an intracellular enzyme responsible for the production of NO. It is produced in cells via the process of the methylation of arginine residues of nuclear proteins. Under normal circumstances, most ADMA is degraded within the cytosol of the cell in which it was formed. In situations of excessive intracellular concentration, some of the ADMA exits the cell into the bloodstream so that it can be degraded in a cell other than the one in which it was formed and only 20% of circulating ADMA is excreted in the urine [16]. All CV risk factors known so far, as well as CVDs, increase the concentration of ADMA in such a way that they lead to the excessive formation or inhibition of its degradation [17]. We found in the literature that ADMA showed a positive correlation with the incidence of LVDD [18] and N-terminal pro-brain NP (NT-proBNP) values [19], and, similarly to NT-proBNP, can further improve the stratification of the risk of cardiac decompensation, major adverse CV events, and mortality in patients with HF [20,21]. Through our research, we aimed to determine whether we could contribute to a personalized understanding of the (residual) CV risk assessed via the SCORE2 algorithm by measuring the plasma concentration of the biomarkers ADMA and NT-proBNP with dependence on LVDD. The objectives of the study were to examine the associations between the prevalence and severity of LVDD and SCORE2, to examine the association between the occurrence and severity of LVDD and the plasma concentration of ADMA and NT-proBNP, to assess the association of SCORE2 with the plasma concentration of ADMA and NT-proBNP and to determine the effectiveness of ADMA and NT-proBNP in predicting LVDD, HFrEF, and NYHA class III–IV in subjects from the general population.

## 2. Materials and Methods

A cross-sectional population study that included 178 adults (99 men, 79 women), aged 40 to 65 years, who came for a cardiology examination at the Osijek-Baranja County Health Center and Clinical Hospital Center Osijek, was conducted in the period from November 2019 to May 2022. The study complied with the provisions of the Declaration of Helsinki and was approved by the Ethics Committee of Osijek-Baranja County Health Center (ID: 03-939-2/19), Clinical Hospital Center Osijek (ID: R2-7882/2019), and the University “Josip Juraj Strossmayer”, Faculty of Medicine in Osijek (ID: 2158-61-07-19-134). The inclusion criteria were age 40–65 years and willingness to participate in the study was demonstrated by signing a written consent. The exclusion criteria were as follows: malignant disease; acute coronary syndrome; acute infectious disease; congenital heart disease in adults; poor echocardiographic presentation, in which the heart structures were not properly seen; and additionally, in women, the use of oral contraceptives, pregnancy, and being in the 6-month period after childbirth.

Anamnesis data on past illnesses and current symptoms, family history, functions, habits, addictions, and medication intake were taken from all subjects. A basic 12-channel ECG, ambulatory blood pressure (BP), body mass, height, waist, and hip circumference were recorded for each subject. All subjects were given the results of laboratory analysis, which contained the following parameters: red blood count, urea, creatinine, total cholesterol (TC), LDL-C, high density lipoprotein cholesterol (HDL-C), triglycerides, and fasting blood glucose (FBG). From the above data, body mass index (BMI), body surface area (BSA), waist/hip ratio, non-HDL-C, and glomerular filtration rate (eGFR) were calculated using the relevant well-known formulas [9,22,23,24]. To estimate the 10-year risk of fatal and non-fatal CV events, we used the SCORE2 algorithm for high-risk countries (to which Croatia belongs) [9].

To establish the diagnosis of targeted CV risk factors and CVDs, such as dyslipidemia, arterial hypertension (AH), overweight, obesity, metabolic syndrome (MetS), CHD, family history of premature atherosclerotic (ATS) disease, diabetes mellitus (DM), and chronic kidney disease (CKD) of stage 3 or higher, we used criteria in accordance with current guidelines and recommendations of the ESC, the American Heart Association (AHA), and the International Society of Nephrology, as well as reports by WHO [9,25,26,27,28,29,30,31]. When discussing dyslipidemia, we mean values of TC greater than 5 mmol/L, LDL-C greater than 2.6 mmol/L, HDL-C of less than 1.2 mmol/L in women and less than 1 mmol/L in men, non-HDL-C of greater than 3.3 mmol/L, or triglyceride values greater than 1.7 mmol/L, as well as the use of hypolipemic drugs [9,25]. By AH, we mean values of SBP greater than 139 mmHg and diastolic BP (DBP) values greater than 89 mmHg, as well as taking antihypertensive drugs [26]. By overweight, we mean a BMI greater than 25 kg/m^2^, while values of 30 kg/m^2^ and more were defined as obesity [27,28]. The diagnosis of MetS was made in situations where, in addition to abdominal obesity (waist circumference greater than 88 cm in women or greater than 102 cm in men), at least two of the following four criteria were present: 1. impaired fasting glycaemia (IFG), defined by FBG values greater than 5.5 mmol/L, 2. SBP greater than 129 mmHg, 3. DBP greater than 84 mmHg, and 4. dyslipidemia [28,29]. We considered physical inactivity as physical activity of moderate intensity with a total weekly duration of less than 150 min [30]. The diagnosis of CHD was established by reviewing the existing medical documentation, based on functional non-invasive tests for the detection of myocardial ischemia during exercise or by showing the anatomy of the coronary arteries through CT angiography or invasive coronary angiography [31]. By a positive family history of premature ATS disease, we mean illness or death related to ATS in first-degree relatives under 55 years of age in men and 65 years of age in women [32]. DM was diagnosed in subjects who met one of the following criteria: 1. FBG greater than 6.9 mmol/L, 2. blood glucose 2 h postprandial or at any time greater than 11 mmol/L, 3. value of hemoglobin A1c (HbA1c) greater than 6.5%, 4. taking oral antidiabetic drugs, or 5. insulin [33]. Chronic kidney disease (CKD) stage 3 or higher was established in subjects with an eGFR of less than 60 mL/min/1.73 m^2^ [34].

For the two-dimensional TTE examination we used Phillips Affiniti 30 ultrasound (Philips Ultrasound Inc, Bothell, WA, USA) with a 2–4 MHz multifrequency probe. Two-dimensional TTE was carried out in all subjects, using the M-mode, two-dimensional (2D)-mode, pulsed-wave (PW), continuous-wave (CW), and tissue Doppler imaging (TDI), respecting the current guidelines of the American Society of Echocardiography (ASE) and the European Association of Cardiovascular Imaging (EACVI) from 2016 [8]. In assessing the diastolic function of the LV, we used the following four variables: 1. the ratio of the maximum LV filling velocities during the early (E) and late phases of diastole (during atrial contraction [A]), obtained using a PW Doppler positioned over the tips of the mitral valves, the so-called E/A ratio; 2. E/e′ (where e′ represents the maximum speed of early relaxation of the myocardium obtained by TDI positioned over the septal part of the mitral annulus); 3. tricuspid regurgitation systolic peak velocity (TR v max); and 4. the left atrium volume index (LAVI) obtained by the biplane area-length method. If only two parameters were available for evaluation, then we utilized the ratio of the maximum flow velocity over the upper right PV during systole (S) and diastole (D), the so-called S/D ratio, and the difference in the duration of atrial contraction measured over the upper right PV (a dur) and over the tips of the mitral valves (A dur), the so-called a dur–A dur, as an additional parameter. 

Based on the estimated LV diastolic function (LVDF), we further divided the subjects into four numerically equal groups according to the following criteria: 

1. Normal LVDF was established if half or more variables were within normal values: E/A 0.8–2, E/e′ septal ˂ 10, TR v max < 2.8 m/s, LAVI ˂ 34 mL/m^2^, S/D ratio ≥ 1, a dur–A dur ˂ 30 msec

2. LVDD grade 1 was established if half or more variables were within these values: E/A ≤ 0.8 (with E ≤ 50 cm/s), E/e′ septal < 10, TR v max < 2.8 m/s, LAVI ≤ 34 mL/m^2^, S/D ratio ≥ 1, a dur–A dur ˂ 30 msec

3. LVDD grade 2 was established if half or more variables were within these values: E/A 0.8–2, E/e′ septal 10–14, TR v max > 2.8 m/s and LAVI > 34 mL/m^2^, S/D ratio ˂ 1, a dur–A dur ≥ 30 msec and

4. LVDD grade 3 was established if half or more variables were within these values: E/A > 2, E/e′ septal > 14, TR v max > 2.8 m/s, LAVI > 34 mL/m^2^, S/D ratio ˂ 1, a dur–A dur ≥ 30 msec.

The laboratory analysis of ADMA and NT-proBNP was carried out at the Clinical Institute for Laboratory Diagnostics of the Clinical Hospital Center Osijek. It was performed exactly according to the manufacturer’s protocol.

ADMA was determined using the ADMA ELISA reagent (BioVendor Group, Brno, Czech Republic) on the Etimax 3000 ELISA processor (DiaSorin S.p.A, Saluggia (Vercelli), Italy). For its determination, 3 mL of venous blood was sampled in a test tube with ethylendiaminetetraacetic acid (EDTA)-anticoagulant. The sample was then centrifuged for 10 min at 3500 rpm and aliquoted into two 300 µL aliquots. Aliquots were then stored at −70 degrees Celsius until analysis. 

NT-proBNP was determined with the Roche NT-proBNP reagent (Roche Diagnostics GmbH, Mannheim, Germany) on the immunochemical analyzer Cobas 6000 (Roche Diagnostics GmbH, Mannheim, Germany) using the ECLIA method. For the determination of NT-proBNP, an additional 3 mL of venous blood was sampled in a test tube without additives. The sample was left for 30 min until the clotting process was completed and then centrifuged for 10 min at 3500 rpm. Immediately afterwards, NT-proBNP was determined in the serum supernatant.

The diagnosis, phenotype, and severity of HF were confirmed based on symptoms, signs, resting ECG, TTE, and the plasma concentration of NT-proBNP, according to ESC Guidelines for HF, published in 2021 [4].

### Statistical Analyses

To observe a mean effect in the difference of numerical variables between four independent groups of subjects, with a significance level of 0.05 and a power of 0.85, the calculated minimum required sample size was 164 subjects (G*Power, 3.1.2). Categorical data were represented by absolute and relative frequencies. Numerical data were described by the arithmetic mean and standard deviation in the case of distributions that followed the normal one, and in other cases by the median and the limits of the interquartile range (IQR). Differences in categorical variables were tested using the chi-square (χ^2^) test and, if necessary, with Fisher’s exact test. The normality of the distribution of numerical variables was tested with the Shapiro–Wilk test. Differences of normally distributed numerical variables between two independent groups were tested with Student’s *t*-test, and in the case of deviation from normal distribution with the Mann–Whitney U test. Differences of normally distributed numerical variables in the case of three or more independent groups were tested using analysis of variance (ANOVA) and, in cases of deviation from the normal distribution, via the Kruskal–Wallis test. Using logistic regression, we evaluated the influence of plasma concentrations of ADMA and NT-proBNP on the probability of occurrence of LVDD, HFrEF, and NYHA class III–IV and high/very high SCORE2. Receiver operating characteristic (ROC) analysis [35] was applied to determine the optimal threshold value, area under the ROC curve (area under the curve, AUC), and specificity and sensitivity of the tested parameters. The association of normally distributed numerical variables was assessed by Pearson’s correlation coefficient r and, in cases of deviation from the normal distribution, by Spearman’s correlation coefficient ρ (rho). All *p* values were two-sided, and the significance level was set at Alpha = 0.05. The statistical program MedCalc^®^ Statistical Software version 20.215 (MedCalc Software Ltd., Ostend, Belgium; https://www.medcalc.org; accessed on 5 February 2023) was used for data analysis.

## 3. Results

### 3.1. General, Clinical, and Laboratory Characteristics and the Prevalence of CV Risk Factors of the Subjects in the Study Groups

The study was conducted on 178 patients (99 male, 79 female), with a median age of 61 years and an IQR of 49 to 65 years. The most common reasons for referral to cardiology examination were dyspnea (65%), unregulated hypertension (48%), palpitations (35%), angina (32%), positive family history of premature ATS disease (29%), and FH or markedly elevated TC or LDL-C (15%). After we estimated LVDF, subjects were divided into four numerically equal groups (Table 1).

Table 1 shows that there were slightly more men in the sample (56%), who had LVDD grade 2 and 3 (*p* = 0.001) at significantly higher rates compared to women. The median age of the subjects increased with the severity of LVDD (*p* < 0.001). The subjects in the LVDD grade 3 group had a significantly higher prevalence of atrial fibrillation (AF), AH, obesity, CHD, DM (*p* < 0.001), CKD stage ≥ 3 (*p* < 0.001), and IFG (*p* = 0.001). They also had a significantly higher prevalence of diagnosed dyslipidemia (*p* = 0.04), but with lower blood lipid values compared to the subjects in the other groups. The reason for this was a significantly higher percentage of the use of statins. There were no significant differences between the groups in the prevalence of overweight, MetS, physical inactivity and active smoking.

### 3.2. Echocardiographic Parameters in the Study Groups

Regarding the TTE examination, Table 2 shows that the group with LVDD grade 3 had the statistically significantly lowest values of LVEF and the highest LV end-diastolic diameters (LVEDd) and LV end-systolic diameters (LVESd), as well as the highest LV mass index compared to all other groups (*p* < 0.001). Other parameters were within the expected values with respect to the estimated LVDF.

### 3.3. Differences between Groups Regarding the Prevalence of HF, NYHA Class, and SCORE2

Table 3 shows a statistically significant difference in the prevalence of HF among all groups, i.e., in the group with normal LVDF, not a single case of HF was recorded, therefore it was not applicable to determine the phenotype and severity of HF in that group, while in all other groups, the prevalence was high and the grade of LVDD tended to be more severe (*p* < 0.001). As for SCORE2, most patients, as many as 104 (58%) had very high CV risk, more significantly in the LVDD grade 2 and 3 groups (*p* < 0.001). 

### 3.4. Differences between Groups Regarding Therapy

A total of 152 (85%) respondents were taking drugs from the antihypertensives, angiotensin receptors and neprilysin inhibitors (ARNIs), diuretics, mineralocorticoid receptor antagonists (MRAs), hypolipemics, antidiabetics, antiarrhythmics, antianginal, antiplatelet and anticoagulant drug groups. The percentage of medication intake within each group depended on the prevalence of CV risk factors, CVDs, HF, and estimated SCORE2 and was therefore highest in the LVDD grade 2 and 3 groups, where all subjects (100%) were taking some form of medication from the previously mentioned groups, often in combination. Using the χ^2^ test, we assessed differences between groups regarding therapy intake. Subjects in the LVDD grade 3 group took the following groups of drugs significantly more often (*p* < 0.001): angiotensin converting enzyme inhibitors (ACEI), ARNI, beta blockers (BB), diuretics, MRAs, statins, amiodarone, trimetazidine, sodium-glucose co-transporter 2 (SGLT2) inhibitors, warfarin, and non-vitamin K antagonist oral anticoagulants (NOACs).

### 3.5. Biomarkers of HF and ED

The median concentration of NT-proBNP of all subjects was 139 ng/L (IQR from 41 ng/L to 680 ng/L), and the median concentration of ADMA was 0.436 µmol/L (IQR from 0.231 µmol/L to 0.528 µmol/L).

In Table 4, we demonstrate the distribution of subjects depending on the observed parameters (LVDD, SCORE2, HF, and NYHA class) and the plasma concentrations of ADMA and NT-proBNP. The lowest values of ADMA (up to 0.329 µmol/L = 1st tertile) had subjects with LVDD grades 2 and 3, very high SCORE2 (*p* < 0.001), and HF (*p* ˂ 0.001), while the highest values (≥0.510 µmol/L = 3rd tertile) were recorded significantly more often in subjects with normal LVDF and LVDD grade 1 (*p* < 0.001). In the case of NT-proBNP, the lowest values (up to 63 ng/L = 1st tertile) had subjects with normal LVDF and LVDD grade 1, while the highest values (≥344 ng/L = 3rd tertile) were more often found in subjects with LVDD grades 2 and 3, very high SCORE2 (*p* < 0.001), HF (predominantly HFrEF (*p* < 0.001)), and NYHA class III-IV (*p* = 0.001).

Using Fisher’s exact test and the χ^2^ test, we compared the concentration of NT-proBNP and ADMA with the general and laboratory characteristic of subjects, as well as the prevalence of CV risk factors listed in Table 1. The highest values of NT-proBNP in plasma were observed in men; subjects aged over 60 years; those with AH, DM, or CKD stage ≥ 3; CHD (*p* = 0.001); and IFG (*p* = 0.006), while the lowest values had the subjects with higher values of TC and non-HDL-C (*p* < 0.001), lower values of FBG (*p* = 0.01), and a positive family history of premature ATS disease (*p* = 0.02). Regarding ADMA, the highest values in plasma belonged to the subjects with higher values of LDL-C (*p* = 0.002), non-HDL-C (*p* = 0.007), and TC (*p* = 0.01), while the lowest plasma values belonged to subjects aged over 60 years and those with AH, IFG, (*p* < 0.001) and CHD (*p* = 0.002). Other CV risk factors listed in Table 1, as well as ambulatory measured SBP and DBP, did not have a statistically significant effect on NT-proBNP or ADMA values in the observed population. There was no statistically significant difference in ADMA concentrations between genders.

Finally, using Fisher’s exact test, we compared the values of the aforementioned biomarkers with the type of drugs used. The subjects who used ACEIs, BBs, statins, insulin, acetylsalicylic acid (ASA), ARNI, MRAs, SGLT2 inhibitors, and diuretics significantly more often had the lowest values of ADMA and highest values of NT-proBNP in plasma. The subjects who used antiarrhythmics, antianginal drugs, NOACs, warfarin, angiotensin receptor blockers (ARBs), and P2Y12 inhibitors were statistically significantly more likely to also have NT-proBNP values in third tertile.

Using logistic regression, we assessed whether plasma concentrations of NT-proBNP and ADMA predict the prevalence of any grade of LVDD (compared to normal LVDF), HFrEF, NYHA class III–IV, and a high/very high SCORE2. Table 5 shows that subjects with higher NT-proBNP values were more likely to have any grade of LVDD, HFrEF, NYHA class III–IV, and high/very high SCORE2. In contrast, the subjects who had lower ADMA values were more likely to have any grade of LVDD and high/very high SCORE2, while ADMA did not show statistical significance according to prevalence of HFrEF and NYHA class III–IV.

As can be concluded from Table 5, we found low but significant odds ratio (OR) values combined with a very narrow confidence interval (CI) of 95%, and therefore, we also performed ROC analysis to determine the optimal threshold value, the area under the ROC curve (AUC), and the specificity and sensitivity of the tested parameters. All the ROC curves demonstrate the superiority of NT-proBNP over ADMA (Figure 1, Figure 2, Figure 3 and Figure 4).

Figure 1, Figure 2 and Figure 3 show the ROC curve comparing NT-proBNP and ADMA plasma concentrations in different groups of subjects depending on the estimated LVDF.

Figure 4 shows the ROC curve comparing NT-proBNP and ADMA plasma concentrations in the subjects with a high/very high SCORE2. This shows that an estimated high/very high SCORE2 is a good diagnostic indicator for low ADMA and high NT-proBNP values (*p* < 0.001).

### 3.6. Medical Therapy and ADMA Values

We determined the connection between therapy and ADMA values. Out of a total of 152 subjects, 33 (22%) were undergoing monotherapy and 119 (78%) were undergoing polytherapy. We found that the subjects who took drugs from the groups listed in Table 6 had lower plasma ADMA values. Although the correlations were weak (r < −0.5), they were significant. The strongest correlation was among those taking ARNI (r = −0.454), ACEIs (r = −0.418), and statins (r = −0.395). They had significantly lower plasma ADMA values compared to those who were not taking drugs from these groups. The situation was the same in those who were undergoing polytherapy—they had significantly lower ADMA values compared to those undergoing monotherapy (r = −0.431).

Using multivariate linear regression, we determined that two predictors were significant for explaining the value of ADMA: polytherapy (β = −0.142) and statins (β = −0.092), which explain 22.3% of the total variance of ADMA (F_(2, 175)_ = 26.5, *p* < 0.001). 

## 4. Discussion

The key findings of this study are as follows. First, certain groups of drugs, and more strongly, in combination, significantly affect the lowering of circulating ADMA concentration in plasma. Second, the same groups of drugs did not have the same effect regarding lowering plasma NT-proBNP concentration. Third, in contrast to previously published studies [15,18,19], we did not confirm a positive correlation of plasma ADMA concentration with NT-proBNP or the prevalence and severity of LVDD, HF, NYHA class, or CV risk, probably due to the previously mentioned drug influence. Fourth, similarly to other studies, we also confirmed that there is a positive correlation between the prevalence and severity of LVDD, CV risk factors, CVDs [5,6,7], estimated CV risk [36], NT-proBNP concentration [37], and HF [38,39]. However, contrary to other authors, we used SCORE2 to assess CV risk. Fifth, we confirmed that a higher concentration of NT-proBNP in the plasma is able to predict the severity of HF and NYHA class III–IV, which was also previously reported by other authors [40,41,42].

In the following section, we will refer only to the results that we obtained in a different manner to those previously published and attempt to justify our opinions, primarily related to the biomarkers that we used in our study.

It is already known that coronary ED is a condition involved not only in vasoconstriction, inflammation, fatty streak formation, and progression to plaque rupture and coronary artery thrombosis but also in the development and progression of LV hypertrophy and LVDD, ultimately resulting in HF development [11,12,43,44,45,46]. Moreover, Reriani et al. [47] found that measuring coronary ED provided better prognostic results compared to the Framingham risk score. It is also known that if risk factors are treated appropriately, coronary and systemic ED are reversible conditions, suggesting that ED becomes a surrogate endpoint of a therapeutic approach to reducing CV risk and CVD incidence [43,44]. However, the gold standard for the evaluation of coronary ED is the angiographic evaluation of vasodilation responses to the administration of vasoactive substances in the coronary arteries, but due to its invasiveness and cost, it is used only in patients who require cardiac catheterization for indications other than ED evaluation. The FMD of the brachial arteries provides a non-invasive alternative but is technically challenging and requires extensive training and standardization [48]. Therefore, it is not surprising that in recent decades there has been a great demand for biomarkers of ED that can be measured from a plasma sample with minimal invasiveness, which would be simple for daily use and inexpensive and accessible for a large number of patients. So far, a large number of such biomarkers have been found, and each of them may have important implications, but also limitations, in the clinical setting, which is why their use is still scientific in terms of researching additional information regarding the risk of developing CVDs and new treatment targets [49]. 

In our study, we used ADMA as a biomarker of ED and NT-proBNP as a biomarker of HF. ADMA is one of the most potent endogenous inhibitors of the three isoforms of NOS that have been related to many CV risk factors and a wide range of CV and other diseases [50]. In 2017, Nemeth, B., et al. [51] published a systematic review and meta-analysis based on which they estimated the reference values of ADMA in a healthy population. The meta-analysis included a total of 5528 adults without hypertension, diabetes, or obesity, who were not taking any medications and were included in one of 66 studies that examined plasma ADMA levels. In 24 of these studies, encompassing a total number of 1435 respondents, the ELISA method was used and reference values of 0.25–0.92 µmol/L were obtained, with an average value of 0.57 µmol/L. In a study published by Deneva-Koycheva et al. [52], the reference range for plasma ADMA concentration using an ELISA method in a sample of 150 healthy Bulgarian residents (74 men and 76 women, aged between 18 and 65 years) were from 0.22 to 0.69 µmol/L, with a mean concentration of 0.48 µmol/L. In the same study, another important fact was established—that there were no gender and age differences in ADMA concentration. Similarly, in our study, the range of ADMA concentration in plasma, also using the ELISA method, was between 0.23 µmol/L and 0.53 µmol/L, with a median of 0.44 µmol/L. Contrary to the previously mentioned study, the concentration of ADMA in our subjects was inversely proportional to age. However, we must emphasize that our study included a general population in which only a few subjects were healthy, while a large percentage had been diagnosed with one of the traditional CV risk factors and CVDs, the prevalence of which increased with age, which is why even 85% of respondents took drugs from the antihypertensive, antidiabetic, hypolipemic, antianginal, antiarrhythmic, anticoagulant, or antiplatelet therapy groups. Those who were taking drugs, especially combinations of drugs from several groups, had statistically lower values of ADMA concentration in plasma (often lower than the specified reference range) compared to the other subjects. In previous reviews and meta-analyses, it has been confirmed that elevated plasma ADMA concentration correlates with the presence and functional significance of plaque in the coronary, cerebral, and peripheral circulation and, consequently, is able to predict CV mortality and morbidity [53,54]. From this, we can conclude that an elevated level of ADMA is associated with an increased CV risk. However, cross-sectional studies that compared ADMA values in the plasma of patients with confirmed CHD in dependence to number and percentage of stenosis of coronary arteries affected by atherosclerosis gave conflicting results. While Mangiacapra et al. [55] confirmed that the level of ADMA in the plasma of 281 patients with CHD is an independent predictor of the expansion and functional significance of atherosclerotic CHD, Ghayour-Mobarhan M et al. [56] did not confirm a significant relationship between the concentration of ADMA in plasma and the presence or severity of coronary artery stenosis in a sample of 165 subjects without traditional CV risk factors. What is not known, regarding both studies, is whether the effect of drug therapy on ADMA concentration was examined. Although a specific ADMA-lowering agent has not yet been found, many drugs have been reported to lower ADMA levels in clinical studies by either increasing the activity/expression of the enzyme responsible for ADMA degradation or by decreasing the expression of the enzyme responsible for the formation of ADMA in cells or via some other unknown method [17,43,44,57,58]. Some of these are ACEIs, ARBs, BBs, hipolipemics, oral hypoglycemics, and ASA. In our study, subjects with AH, IFG, and CHD, who mostly belonged to the groups with LVDD grades 2 and 3, in which the percentage of subjects who were taking one or more drugs from the groups mentioned above was 100%, had the lowest plasma ADMA values, while subjects with dyslipidemia had the highest plasma ADMA values. Although dyslipidemia was the most common CV risk factor in the population we observed, with a total prevalence of 92%, statins were taken by only 45% of respondents and were statistically most significantly used in the LVDD stage 3 group (even 81%). Therefore, we believe that the obtained results are the result of the drugs, and we confirmed this through multivariate regression, according to which, the variations of ADMA values in plasma are most significantly affected by polytherapy and statins.

NT-proBNP is the second biomarker that we examined in our study. It is a biologically inactive N-terminal fragment that is formed from the cleavage of prohormone (proBNP) into the biologically active hormone BNP and is produced by LV myocardial cells in response to myocardial stretching under the influence of increased pressure and/or volume. NT-proBNP, as well as BNP, has an important place not only in the diagnosis of HF [59] but also in the prevention of both HF and major adverse CV events [60,61]. It has been confirmed in several studies so far [40,62] that the value of NT-proBNP is significantly higher in women compared to men at any age, while the levels increase with age in both sexes. Although, in our study, we also confirmed that NT-proBNP concentration increased with age, men had higher values of plasma NT-proBNP concentrations compared to women. We explain this by the fact that there were slightly more men in the sample, who had significantly higher rates of LVDD grades 2 and 3 compared to women, and the higher LVDD was, the higher NT-proBNP concentrations in plasma was. In terms of current knowledge concerning the effects of drugs on the concentration of NPs in plasma, theories are quite contradictory. Recent studies confirm the reduction of NT-proBNP with the use of new drugs in the treatment of HFrEF—while ARNI lowers the initial values of NT-proBP by approximately 30%, SGLT2 inhibitors and vericiguat lower these values more modestly (by about 10–15%) [63]. The effects of other drugs that we are used in daily practice for HF, CHD, and AH are more complex and depend not only on the underlying disease for which the given drug is introduced but also on the length of therapy as well as on the active substance within a particular group of drugs [64,65]. 

We believe that our study has important implications for clinical practice, indicating that the therapy we use today can have a great impact on the improvement of ED, and if introduced early enough, it can influence the sequence of the development of CVDs and HF. In this context, ADMA can serve as a good biomarker of treatment effectiveness.

The main limitation of our study lies in the fact that we conducted a cross-sectional study in which we compared the values of NT-proBNP and ADMA between different groups of people depending on estimated LVDF, and not, as in previous prospective studies, comparing the same groups of people after a certain period of undergoing certain therapies. Therefore, we cannot say with certainty how significant each therapy is in lowering NT-proBNP and ADMA, because we did not monitor these values over time. Therefore, a longitudinal study is highly needed.

## 5. Conclusions

In the pathophysiological continuum of CVDs, the first link is ED, caused by traditional and non-traditional CV risk factors, which eventually lead to ischemia, myocardial remodeling, and finally, to HF. The biomarkers of ED can be measured from a plasma sample with minimal invasiveness, they are suitable for daily use, inexpensive and accessible to a large number of patients. ADMA, similar to other biomarkers of ED, gives us information about the systemic state of endothelial function, and with the help of LVDF assessment and measurement of NT-proBNP concentration in plasma, we are able to non-invasively assess the coronary consequences of ED. Further research should determine how the biomarkers of ED, LVDD, and NT-proBNP could be included in the CV risk assessment tables. It is also necessary to clarify the knowledge regarding which period of life it becomes advisable to start measuring and how often to examine the biomarkers of ED and what the target values would be, with the aim of slowing the progression of the development of CVDs, which could ultimately result in a decrease in the incidence of CVDs and HF. Furthermore, a longitudinal study is needed to determine the effect of therapy on the plasma values of NT-proBNP and ADMA.

## Figures and Tables

**Figure 1 ijerph-20-04433-f001:**
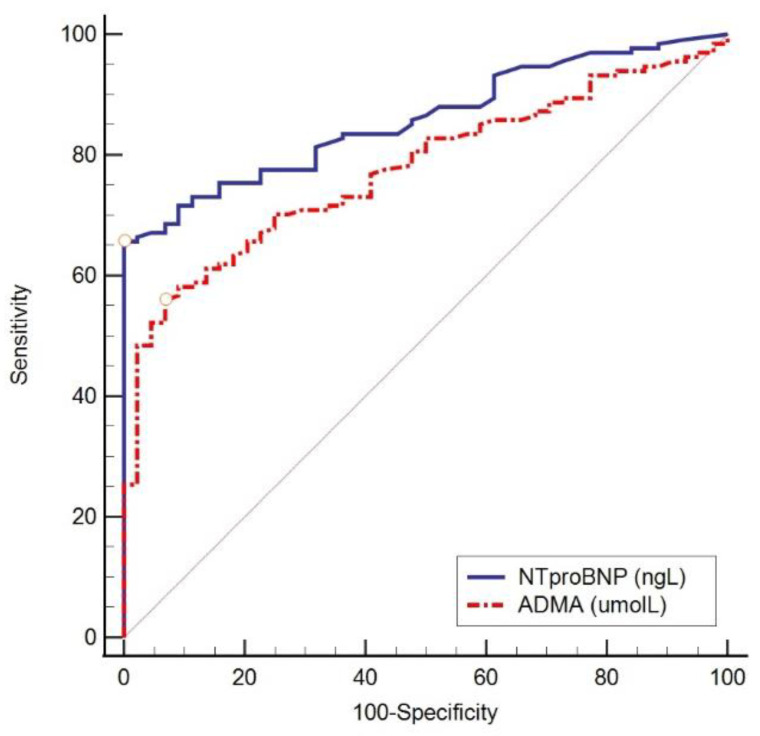
ROC curve comparing NT-proBNP and ADMA in different groups of subjects with normal LVDF and any form of LVDD. NT-proBNP was a better diagnostic indicator of the existence of any grade of LVDD (AUC of 0.859, 95% CI 0.807–0.912, sensitivity 65.7%, specificity 100%, cut-off values > 141 ng/L, Youden index 0.66, *p* < 0.001) compared to ADMA (AUC 0.777, 95% CI 0.708–0.845, sensitivity 56%, specificity 93.2%, cut-off values ≤ 0.392 µmol/L, Youden index 0.49, *p* < 0.001).

**Figure 2 ijerph-20-04433-f002:**
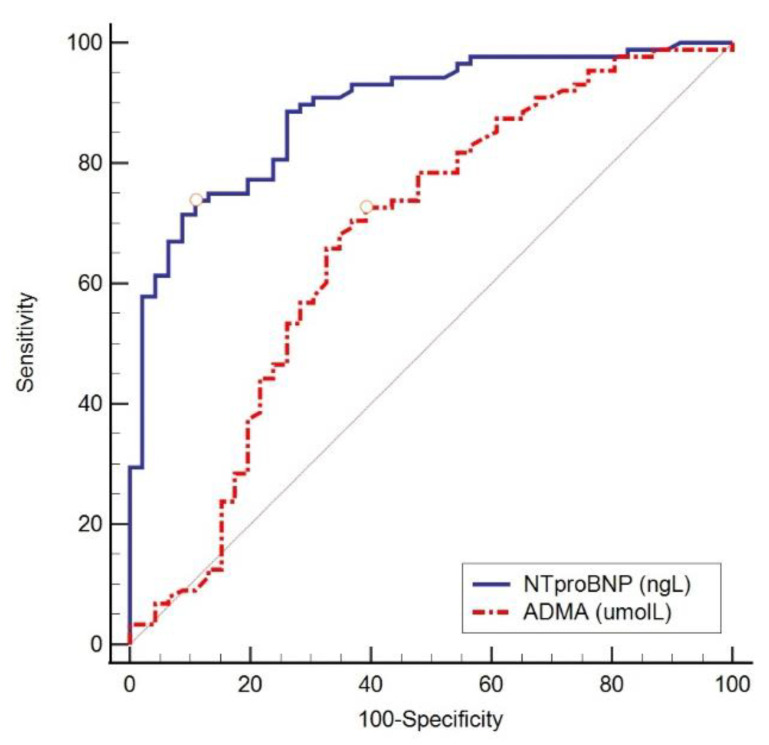
ROC curve comparing NT-proBNP and ADMA in different groups of subjects with LVDD grade 1 and LVDD grades 2 and 3. NT-proBNP was a better diagnostic indicator of the existence of LVDD grades 2 and 3 (AUC 0.889, 95% CI 0.833–0.945, sensitivity 73.9%, specificity 89.1%, cut-off values > 220 ng/L, Youden index 0.63, *p* < 0.001) compared to ADMA (AUC 0.673, 95% CI 0.570–0.777, sensitivity 72.7%, specificity 60.9%, cut-off values ≤ 0.433 µmol/L, Youden index 0.34, *p* = 0.001).

**Figure 3 ijerph-20-04433-f003:**
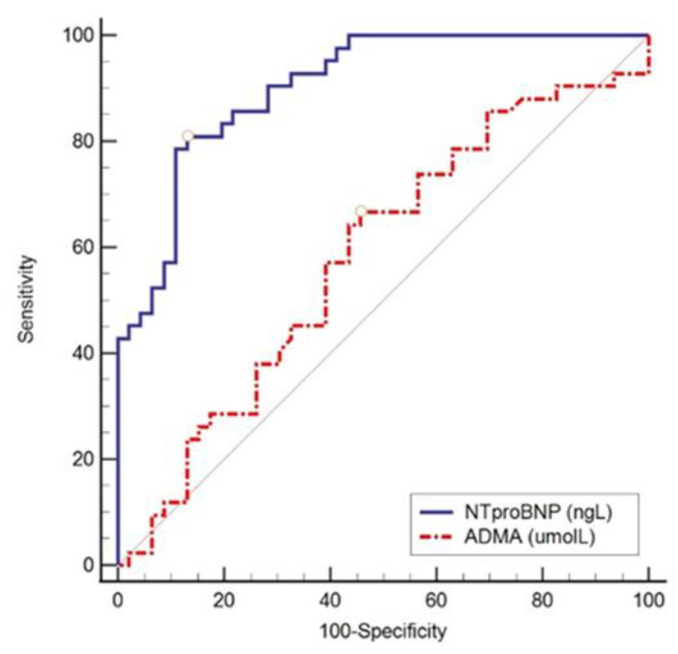
ROC curve comparing NT-proBNP and ADMA in different groups of subjects with LVDD grade 2 and LVDD grade 3. NT-proBNP again proved to be a significant parameter in differentiating LVDD grade 2 from LVDD grade 3 (AUC 0.904, 95% CI 0.844–0.964, sensitivity 81%, specificity 87%, cut-off values > 784 ng/L, Youden index 0.68, *p* < 0.001), while ADMA did not show statistical significance in differentiating LVDD grades 2 and 3 (AUC 0.582, 95% CI 0.461–0.703, sensitivity 67%, specificity 54%, cut-off values > 0.24 µmol/L, Youden index 0.21, *p* = 0.18).

**Figure 4 ijerph-20-04433-f004:**
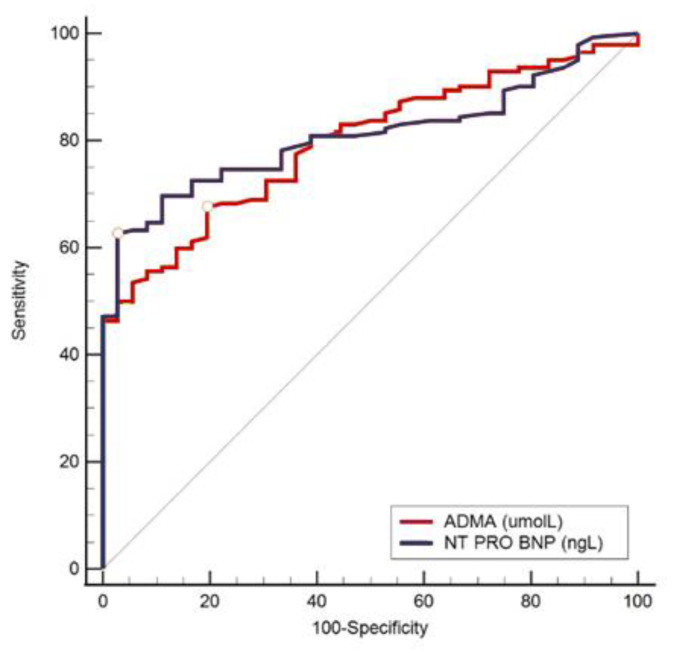
ROC analysis in predicting the laboratory values of ADMA and NT-proBNP using more clinically accessible parameters used for SCORE2. ADMA used the following parameters: AUC 0.796, 95% CI 0.730–0.853, sensitivity 97.6% and specificity 80.6% at cut-off values ≤ 0.475 µmol/L, Youden index 0.482, *p* < 0.001. NT-proBNP had the following parameters: AUC 0.810, 95% CI 0.744–0.865, sensitivity 62.7% and specificity 97.2% at cut-off values > 0.136 ng/L, Youden index 0.599, *p* < 0.001.

**Table 1 ijerph-20-04433-t001:** General, clinical, and laboratory characteristics and prevalence of CV risk factors of the subjects in the study groups.

Number (%) Subjects					
	Normal LVDF (n = 44)	LVDD Grade 1 (n = 46)	LVDD Grade 2 (n = 46)	LVDD Grade 3 (n = 42)	*p*-Value
**General Characteristics**					
Sex					**0.001** *
Male	18 (41)	19 (41)	33 (72)	29 (69)	
Female	26 (59)	27 (59)	13 (28)	13 (31)	
Age (y)					**<0.001** ^†^
Median of age	44	57	62	65	
Interquartile range	42–51	50–64	59–64	61–65	
**Anthropometric Characteristics (Median (IQR))**					
BMI (kg/m^2^)	27.1 (24.3–28.9)	28.2 (25.5–31.3)	29.4 (26.6–32.4)	29.8 (25–33.4)	0.06 ^†^
BSA (m^2^)	1.9 (1.8–2.2)	2 (1.7–2.2)	2.1 (1.9–2.3)	2.1 (1.9–2.2)	0.26 ^†^
Waist circumference (cm)	94.5 (86–102)	97 (85–107)	100 (95–108)	104 (97–112)	**0.002** ^†^
Hip circumference (cm)	100 (95–109)	106 (98–114)	110 (104–119)	109 (102–118)	**<0.001** ^†^
Waist/hip ratio	0.9 (0.9–1)	0.9 (0.9–1)	0.9 (0.9–1)	0.9 (0.9–1)	0.20 ^†^
**Blood Pressure (Median (IQR))**					
Systolic	130 (120–135)	139 (130–142)	135 (120–140)	130 (119–146)	0.09 ^†^
Diastolic	80 (80–88)	80 (78–90)	80 (80–90)	80 (71.5–90)	0.98 ^†^
**ECG**					
Median frequency (bpm) (IQR)	75 (61–85)	71 (62–79)	67 (60–80)	87 (74–99)	**<0.001** ^†^
AF (n (%)) patients	0	2 (4)	19 (41)	33 (79)	**<0.001** *
**Laboratory Analysis (Median (IQR))**					
Urea (mmolL)	5.2 (4.3–6.2)	6 (5.3–7)	6.3 (5.1–8)	7.1 (5.7–8.9)	**<0.001** ^†^
Creatinine (µmolL)	72 (64–82)	74 (64–82)	82 (73–95)	99 (75–121)	**<0.001** ^†^
eGFR (mL/min/1.73 m^2^)	108 (95–115)	97 (87–109)	88 (73–97)	69 (54–94)	**<0.001** ^†^
TC (mmol/L)	5.8 (5,3–6,4)	5.8 (4.9–6.5)	4.8 (3.9–5.8)	4.6 (3.1–5.5)	**<0.001** ^†^
HDL-C (mmol/L)	1.4 (1.2–1.7)	1.3 (1.1–1.4)	1.2 (1.1–1.3)	1.1 (0.8–1.4)	**0.001** ^†^
non HDL-C (mmol/L)	4.3 (3.7–5.3)	4.5 (3.5–5.1)	3.3 (2.7–4.5)	2.9 (2.2–4.2)	**<0.001** ^†^
LDL-C (mmol/L)	3.6 (3.1–4.2)	3.7 (2.7–4.3)	2.8 (2–3.8)	2.7 (1.8–3.7)	**<0.001** ^†^
Triglycerides (mmol/L)	1.2 (0.9–1.9)	1.7 (1.3–2.3)	1.6 (1.2–2)	1.2 (0.9–1.7)	**0.004** ^†^
FBG (mmol/L)	5.1 (4.7–5.6)	6.2 (5.3–6.7)	5.8 (5.1–6.9)	6 (5.4–7.6)	**<0.001** ^†^
**CV Risk Factors (Number (%) Subjects)**					
Dyslipidemia	37 (84)	41 (89)	44 (98)	41 (98)	**0.04** *
AH	18 (41)	41 (89)	42 (91)	40 (95)	**<0.001** *
Overweight	23 (53)	17 (49)	19 (68)	11 (50)	0.44 *
MetS	19 (43)	25 (54)	24 (52)	29 (69)	0.11 *
Physical inactivity	24 (55)	23 (50)	16 (35)	23 (55)	0.19 *
Obesity	9 (20)	19 (49)	18 (58)	20 (65)	**<0.001** *
CHD	0	14 (30)	29 (63)	27 (64)	**<0.001** *
IFG	4 (9)	13 (35)	15 (44)	13 (46)	**0.001** *
Family history of premature ATS disease	22 (50)	12 (26)	8 (17)	10 (24)	**0.004** *
Active smoking	14 (32)	13 (28)	11 (24)	11 (26)	0.86 *
DM	0	9 (20)	12 (26)	17 (40)	**<0.001** *
CKD stage ≥ 3	0	1 (2)	6 (13)	15 (36)	**<0.001** ^§^

* χ^2^ test; ^†^ Kruskal–Wallis test; ^§^ Fisher’s exact test; LVDF—left ventricular diastolic function; LVDD—left ventricular diastolic dysfunction; IQR—interquartile range; BMI—body mass index; BSA—body surface area; ECG—electrocardiogram; AF—atrial fibrillation; eGFR—estimated glomerular filtration rate; TC—total cholesterol; HDL-C—high density lipoprotein cholesterol; LDL-C—low density lipoprotein cholesterol; FBG—fasting blood glucose; CV—cardiovascular; AH—arterial hypertension; MetS—metabolic syndrome; CHD—coronary heart disease; IFG—impaired fasting glycaemia; ATS—atherosclerotic; DM—diabetes mellitus; CKD—chronic kidney disease. Bold numbers indicate that statistical significance was demonstrated.

**Table 2 ijerph-20-04433-t002:** Echocardiographic parameters in the study groups.

Median (IQR)					
	Normal LVDF (n = 44)	LVDD Grade 1 (n = 46)	LVDD Grade 2 (n = 46)	LVDD Grade 3 (n = 42)	*p*-Value *
E/A	1.1 (0.8–1.3)	0.8 (0.7–0.9)	0.9 (0.7- 1.1)	2.7 (2.2–3)	**<0.001**
E/e′	8 (6.6–9)	12 (10–13.5)	15 (14–17.3)	23 (18.5–28)	**<0.001**
LAVI (mL/m^2^)	21 (17.1–25)	23 (18.8–28)	36 (29–42)	48 (40–63.3)	**<0.001**
LVEDd (cm)	4.7 (4.4–5.2)	4.8 (4.3–5.2)	5 (4.6–5.6)	5.9 (5.2–6.1)	**<0.001**
LVESd (cm)	3.1 (2.9–3.5)	3.3 (2.9–3.6)	3.7 (3.3–4)	4.5 (3.7–5.1)	**<0.001**
LVMI (g/m^2^)	97.5 (80–111.8)	116.5 (93.5–134.5)	140 (108.8–160.8)	149 (127.3–178.3)	**<0.001**
LVEF classes	p	p	p	r	**<0.001**

* Kruskal–Wallis test; LVDF—left ventricular diastolic function; LVDD—left ventricular diastolic dysfunction; E—early phase of diastole; A—atrial contraction (late) phase of diastole; e′—maximum speed of early relaxation of the myocardium obtained by TDI positioned over the septal part of the mitral annulus; LAVI—left atrium volume index; LVEDd—left ventricle end-diastolic diameter; LVESd—left ventricle end-systolic diameter; LVMI—left ventricular mass index; LVEF—left ventricular ejection fraction; p—preserved; r—reduced. Bold numbers indicate that statistical significance was demonstrated.

**Table 3 ijerph-20-04433-t003:** Differences between groups in expression of HF, NYHA class, and SCORE.

Number (%) Subjects						
	Normal LVDF (n = 44)	LVDD Grade 1 (n = 46)	LVDD Grade 2 (n = 46)	LVDD Grade 3 (n = 42)	Total (n = 178)	*p*-Value *
**HF**	**0**	**13 (28)**	**36 (78)**	**42 (100)**	**91 (51)**	**<0.001** *
HFpEF	/	10 (77)	18 (50)	11 (26)	39 (43)	**<0.001** *
HFmrEF	/	2 (15)	13 (36)	7 (17)	22 (24)	**<0.001** *
HFrEF	/	1 (8)	5 (11)	24 (57)	30 (33)	**<0.001** *
**NYHA (n = 91)**	/	**13**	**36**	**42**	**91**	
Class I-II	/	11 (85)	25 (69)	13 (31)	49 (54)	**<0.001** *****
Class III-IV	/	2 (15)	11 (31)	29 (69)	42 (46)	**<0.001** *****
**SCORE2**						
low to moderate	28 (63)	8 (17)	0	0	36 (20)	**<0.001** ^†^
high	14 (32)	12 (26)	6 (13)	6 (14)	38 (21)	**<0.001** ^†^
very high	2 (5)	26 (57)	40 (87)	36 (86)	104 (58)	**<0.001** ^†^

* Fisher’s exact test; ^†^ χ^2^ test; LVDF—left ventricular diastolic function; LVDD—left ventricular diastolic dysfunction; HF—heart failure; HFpEF—heart failure with preserve ejection fraction; HFmrEF—heart failure with mildly reduced ejection fraction; HFrEF—heart failure with reduced ejection fraction; NYHA—New York Heart Association; SCORE2—Systematic Coronary Risk Evaluation2. Bold numbers indicate that statistical significance was demonstrated.

**Table 4 ijerph-20-04433-t004:** Distribution of study subjects depending on LVDD, SCORE2, HF, and NYHA class and tertiles of ADMA and NT-proBNP in plasma.

	ADMA (µmol/L)			*p*-Value	NT-proBNP (ng/L)			*p*-Value
	Tertile 1	Tertile 2	Tertile 3		Tertile 1	Tertile 2	Tertile 3	
Parameter	59 (33)	58 (33)	61 (34)		58 (33)	61 (34)	59 (33)	
**LVDD (n = 178)**								
Normal LVDF (n = 44)	1 (2)	17 (39)	26 (59)	**<0.001** *	30 (68)	14 (32)	0	**<0.001** *
LVDD Grade 1 (n = 46)	12 (26)	14 (30)	20 (43)		23 (50)	20 (43)	3 (7)	
LVDD Grade 2 (n = 46)	28 (61)	12 (26)	6 (13)		5 (11)	24 (52)	17 (37)	
LVDD Grade 3 (n = 42)	18 (43)	15 (36)	9 (21)		0	3 (7)	39 (93)	
**SCORE (n = 178)**								
Low to medium (n =36)	0	13 (36)	23 (64)	**<0.001** *	24 (67)	12 (33)	0	**<0.001** *
High (n = 38)	7 (18)	12 (32)	19 (50)		19 (50)	12 (32)	7 (18)	
Very high (n = 104)	52 (50)	33 (32)	19 (18)		15 (14)	37 (36)	52 (50)	
**HF (n = 91)**	44 (48)	28 (31)	19 (21)	**<0.001** ^†^	1 (1)	31 (34)	59 (65)	**<0.001** ^†^
HFpEF (n =39)	17 (44)	13 (33)	9 (23)	**0.003** ^†^	0	22 (56)	17 (44)	**<0.001** ^†^
HFmrEF (n = 22)	13 (59)	6 (27)	3 (14)	**<0.001** ^†^	1 (4)	7 (32)	14 (64)	**<0.001** ^†^
HFrEF (n = 30)	14 (47)	9 (30)	7 (23)	**0.003** ^†^	0	2 (7)	28 (93)	**<0.001** ^†^
**NYHA (n = 91)**								
Class I-II (n = 49)	25 (51)	15 (31)	9 (18)	0.76 ^†^	1 (2)	24 (49)	24 (49)	**0.001** ^†^
Class III-IV (n = 42)	19 (45)	13 (31)	10 (24)		0	7 (17)	35 (83)	

* χ^2^ test; ^†^ Fisher’s exact test; ADMA—asymmetrical dimethylarginine; NT-proBNP—N-terminal pro-brain natriuretic peptide; LVDD—left ventricular diastolic dysfunction; LVDF—left ventricular diastolic function; SCORE2—Systematic Coronary Risk Evaluation2; HF—heart failure; HFpEF—heart failure with preserve ejection fraction; HFmrEF—heart failure with mildly reduced ejection fraction; HFrEF—heart failure with reduced ejection fraction; NYHA—New York Heart Association. Bold numbers indicate that statistical significance was demonstrated.

**Table 5 ijerph-20-04433-t005:** Logistic regression.

	β	Wald	*p*-Value	OR (95% CI)
**NT-proBNP**				
LVDD any grade	0.01	11.13	**0.001**	1.01 (1.001–1.02)
HFrEF	0.001	19.5	**<0.001**	1.001 (1.001–1.002)
NYHA class III-IV	0.001	9.49	**0.002**	1.001 (1.0–1.002)
High/very high SCORE2	0.008	6.47	**0.01**	1.01 (1.002–1.014)
**ADMA**				
LVDD any grade	−3.7	6.26	**0.01**	0.02 (0.001–0.45)
High/very high SCORE2	−5.06	9.43	**0.002**	0.006 (0.002–0.159)

β—regression coefficient; OR—odds ratio; 95% CI—95% confidence interval; NT-proBNP—N-terminal pro-brain natriuretic peptide; LVDD—left ventricular diastolic dysfunction; HFrEF—heart failure with reduced ejection fraction; NYHA—New York Heart Association; SCORE2—Systematic Coronary Risk Evaluation2; ADMA—asymmetrical dimethylarginine. Bold numbers indicate that statistical significance was demonstrated.

**Table 6 ijerph-20-04433-t006:** Correlation between ADMA and certain groups of drugs.

	Correlation Coefficient r *	*p*-Value
	ADMA	
**ACEIs**	−0.418	**<0.001**
**ARNI**	−0.454	**<0.001**
**BBs**	−0.352	**<0.001**
**Diuretics**	−0.230	**0.002**
**MRAs**	−0.299	**<0.001**
**Statins**	−0.395	**<0.001**
**SGLT2 inhibitors**	−0.283	**<0.001**
**Insulin**	−0.245	**0.001**
**ASA**	−0.299	**<0.001**
**Polytherapy vs. monotherapy**	−0.431	**<0.001**

* Point-biserial correlation coefficient; ADMA—asymmetrical dimethylarginine; ACEI—angiotensin converting enzyme inhibitor; ARNI—angiotensin receptors and neprilysin inhibitor; BB—beta blocker; MRA—mineralocorticoid receptor antagonist; SGLT2—sodium-glucose co-transporter 2; ASA—acetylsalicylic acid. Bold numbers indicate that statistical significance was demonstrated.

## Data Availability

Not applicable.

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
