# Peer review of "The Relationship of Left Ventricular Diastolic Dysfunction and Asymmetrical Dimethylarginine as a Biomarker of Endothelial Dysfunction with Cardiovascular Risk Assessed by Systematic Coronary Risk Evaluation2 Algorithm and Heart Failure—A Cross-Sectional Study"

_ijerph, 2023, doi:10.3390/ijerph20054433_

Round 1

Reviewer 1 Report

Reviewer comnents

This manuscript presents some valuable feed data about The relationship of left ventricular diastolic dysfunction and asymmetrical dimethylarginine as a biomarker of endothelial dysfunction with cardiovascular risk assessed by Systematic COronary Risk Evaluation2 algorithm and heart failure - a cross-sectional study. The writing is perfect except some minor defects.

1.       The structure of the article is complete and the data is substantial.

2.       The methods need to be supported by citing articles. Some sentences which are indicated in the manuscript need to be supported by reference.

3.       Include statistical tool name in the method section

Reviewer 2 Report

The current manuscript reported the relationship between left ventricular diastolic dysfunction and symmetrical dimethylarginine as a biomarker of endothelial dysfunction with cardiovascular risk. I have one concern about this study. The authors have made a correlation between these two markers. Did they use this conclusion to make any evaluation of the reality from other group patients (use validation and test sets)?

Reviewer 3 Report

The introduction is very long.

Methods are elaborate but acceptable.

The results raise some questions.

- table 1: for LVDD grade 3, blood lipids are lower compared to grade 0-2, yet table 3 states more dyslipidemia in LVDD grade 3.

- table 2 does not add much. For example blockade of RAAS 'ACE, ARB, ARNI), beta blocking agents, mineralocorticoid receptor antagonists and SGLT-2 inhibitors are the standard for treating heart failure (although not always applied).

- table 3 could be grouped with table 1

- table 4 could be limited to the more clinically applied parameters such as LAVI, é/E, LVMI, LV diameters and LVEF classes (p, mr, r in table 5)

- table 6 and 7 are of value; in table 7, ADMA and NT-proBNP seem to carry the same message.

- the results of the logistic regression should also be tabulated. The low OR (1.001 and 1.01) combined with a very narrow 95%CI are surprising. What does this mean clinically. 

- all the ROC curves demonstrate the superiority of NT-proBNT over ADMA.

- "Values of ADMA and NT-proBNP obtained by ROC analysis in predicting high/very high 407 SCORE2": I would go for the reverse: predicting lab values using more clinically accessible parameters used for SCORE2

- how should we interprete the results in table 8? What is its clinical relevancy?

A longitudinal study is highly needed!

Round 2

Reviewer 3 Report

The manuscript has improved. There are still some issues. Are the numerical labels of the tables mentioned in the text corresponding with the tables themselves?  (example: table 3= Table 5 in line 338?)

What are the "0" in the first column of table 3?

Indicate also if 'Low to moderate', 'high' and 'very high' refers to SCORE2

The following line in table 5 logistic regression still amazes me:

NYHA class III-IV: Odds ratio (95%CI) of 1.001 (1.0 - 1.002). With a lower end of 95%CI at 1.0, one would expect a p value of about 0.05 and not 0.002.
